# Combining a Universal Capture Ligand and Pan-Serotype Monoclonal Antibody to Develop a Pan-Serotype Lateral Flow Strip Test for Foot-and-Mouth Disease Virus Detection

**DOI:** 10.3390/v14040785

**Published:** 2022-04-10

**Authors:** Ming Yang, Dmytro Zhmendak, Valerie Mioulet, Donald P. King, Alison Burman, Charles K. Nfon

**Affiliations:** 1National Centre for Foreign Animal Disease, Canadian Food Inspection Agency, Winnipeg, MB R3E 3M4, Canada; dmytro.zhmendak@inspection.gc.ca; 2The Pirbright Institute, Woking GU24 0NF, UK; valerie.mioulet@pirbright.ac.uk (V.M.); donald.king@pirbright.ac.uk (D.P.K.); alison.burman@pirbright.ac.uk (A.B.)

**Keywords:** pan-serotype, foot-and-mouth disease virus, strip test

## Abstract

Foot-and-mouth disease virus (FMDV) causes FMD, a highly contagious disease of cloven-hoofed animals including cattle, goats, pigs and sheep. Rapid detection of FMDV is critical to limit the devastating economic losses due to FMD. Current laboratory methods for FMDV detection such as virus isolation, real-time reverse transcription PCR and antigen detection enzyme-linked immunosorbent assay (AgELISA) are labor-intensive, requiring trained personnel and specialized equipment. We present the development and validation of a pan-serotype lateral flow strip test (LFST) that uses recombinant bovine integrin αvβ6 as a universal capture ligand and a pan-serotype monoclonal antibody (mAb) to detect FMDV. The LFST detected all seven FMDV serotypes, where the diagnostic sensitivity was comparable to the AgELISA, and the diagnostic specificity was 100% without cross-reactivity to other viruses causing vesicular disease in livestock. This rapid test will be useful for on-site FMDV detection, as well as in laboratories in endemic countries where laboratory resources are limited.

## 1. Introduction

Foot-and-mouth disease (FMD) is caused by FMD virus (FMDV). FMD is acute and highly contagious and affects cloven-hoofed animals such as cattle, goats, pigs and sheep. FMD is a major economic concern for the livestock industry in many developing countries and is a continued threat to countries that are FMD-free because of its potential negative impact on trade in agricultural products. Rapid detection of FMDV is essential for swift control of outbreaks [1,2]. 

Multiple laboratory-based methods are available for FMDV detection, including virus isolation, real-time reverse transcription (rRT) PCR, antigen detection enzyme-linked immunosorbent assay (AgELISA) and genomic sequencing. While some molecular-based methods can be performed in the field such as rRT-PCR and RT loop-mediated isothermal amplification (LAMP) [3,4,5,6], trained personnel and specialized equipment may still be required [7]. 

Rapid lateral flow strip tests (LFSTs) are routinely used for the detection of bioactive molecules (hormones, drugs and toxins) and infectious agents [8,9,10]. They are easy to use, and results can be obtained in 10–30 min. FMDV LFSTs have previously been reported as sensitive and suitable for rapid on-site diagnosis, with sensitivity comparable or better than AgELISAs [11,12,13,14,15,16], and can detect FMDV in tissue homogenates, vesicular fluid, oral fluids and lesion swabs [14,17,18]. 

FMDV enters cells by attaching to an integrin heterodimeric receptor [19], and integrin αvβ6 has been shown to universally bind all FMDV serotypes [20,21,22,23]. Ferris et al. demonstrated recombinant bovine integrin that recognizes FMDV from sheep, goats and other species [21]. The use of recombinant bovine integrin (RBIαvβ6) as a capture ligand can make the diagnosis of FMD simpler since it circumvents the need for multiple polyclonal or monoclonal antibodies [23], but when RBIαvβ6 is used on its own, the reagent does not have the required FMDV specificity for diagnostic use. Therefore, ELISA formats that couple RBIαvβ6 with FMDV-specific monoclonal antibodies have been developed [24]. In this study, we aimed to develop a novel LFST for the detection of all seven serotypes of FMDV using RBIαvβ6 as a capture ligand, and a pan-serotype monoclonal antibody (mAb) [24] as the detection agent.

## 2. Materials and Methods

### 2.1. Production and Biotinylation of Recombinant Bovine Integrin αvβ6 

The transfection and small-scale expression of RBIαvβ6 have previously been described by scientists at the FAO World Reference Laboratory for FMD (WRLFMD), Pirbright Institute, UK [23]. For larger-scale production, confluent HEK293T cells (ATCC, CRL-3216) were grown in expanded surface roller bottles, and the transfection agent PEI (polyethylenimine) was used for cost effectiveness. For each roller bottle, 0.5 mg of DNA (0.25 mg of each plasmid, αv-his-tagged and B6-his-tagged) and 0.875 mL of 1 mg/mL PEI were combined in 25 mL serum-free DMEM and left for 10 min at room temperature. This mix was then added to the roller bottle containing 100 mL DMEM supplemented with 2% FCS. The roller bottle was incubated for 5 to 6 days, then the supernatant was harvested and the his-tagged integrin was purified using a His-Trap FF column (GE Healthcare).

The purified RBIαvβ6 was biotinylated using BiotinTag Micro Biotinylation Kit (Cat# BTAG, Sigma-Aldrich, Burlington, MA, USA). BAC-SulfoNHS was dissolved using 30 µL DMSO and then added to 0.1 M PBS to a final concentration of 5 mg/mL. The dissolved BAC-SulfoNHS (10 µL) was added to the recombinant RBIαvβ6 (175 ug) in 0.1 M PBS and incubated for 30 min at room temperature. Following incubation, the unbound biotin was removed by dialysis against PBS at 4 °C. The biotinylated RBIαvβ6 was stored at 4 °C in PBS with 0.01% NaN_3_.

### 2.2. Purification and Gold Conjugation of the Monoclonal Antibody 

The pan-serotype FMDV monoclonal antibody (mAb F21-42 [24]) was purified using a Hi-Trap Protein-G affinity column (GE, Fairfield, CT, USA) and an AKIA chromatography system. 

The antibody was gold-conjugated using the High Sensitivity Conjugation kit (80 nm Gold Nanospheres and 150 nm Gold Nanoshells, nanoComposix, Inc., San Diego, CA, USA). Briefly, 70 µg 1-Ethyl-3-(3-dimethylaminopropyl)carbodiimide (EDC) and 140 µg Sulfo-NHS were added to 1 mL gold solution and incubated for 30 min at room temperature to activate carboxy gold. The gold solution was washed twice with 1 mL reaction buffer (potassium pH 7.4). The purified mAb F21-42 (20 µg) was added to the gold solution and incubated for 1 h at room temperature, followed by washing with reaction buffer twice. The gold-conjugated mAb F21-42 was re-suspended in the conjugation buffer (PBS with 0.5% BSA, 0.5% casein, 1% Tween-20 and 0.05% NaN_3_) and stored at 4 °C. 

### 2.3. Experimental Samples

The majority of FMDV-positive samples representing different serotypes used in this study were obtained from the WRLFMD (The Pirbright Institute, Woking, UK) and processed as previously described [3,14,15,16,17,18]. FMDV-positive tissues were obtained from previously reported experimental studies in cattle, sheep and pigs [3,14,15,16,17,18] and 10% tissue suspensions prepared as previously described [3,14,15,16,17,18]. Negative epithelial tissues were collected from naïve controls in experimental studies and from abattoirs in Manitoba, Canada.

### 2.4. Development of Pan-Serotype FMD Lateral Flow Strip Test 

FMDV samples (culture supernatants or tissue suspensions, 50 µL) were mixed with the biotin-conjugated RBIαvβ6 (0.5 µL/each) and the gold-conjugated detection mAb F21-42 (2 µL for each 80 nm and 150 nm gold conjugate) to form a complex in a running buffer (reaction mix). 

Readymade gRAD strips (Bioporto Diagnostics A/S, Copenhagen, Denmark) had a biotin–binding protein sprayed on the test line and anti-mouse antibody sprayed on the control line [14,15,16]. The gRAD strips were dipped into the reaction mix which then flowed through the strip by capillary action. The results were determined through visualization after 30 min. A positive result was indicated by bands on both the test and control lines. A negative result was indicated by a single band on the control line only. 

### 2.5. Antigen Detection Enzyme-Linked Immunosorbent Assay

Antigen detection ELISA for each FMDV serotype was performed as previously described [14,15,16,17,18,24].

### 2.6. Real-Time Reverse Transcription Polymerase Chain Reaction

Real-time reverse transcription polymerase chain reaction (rRT-PCR) was performed as previously described [3,18,25].

## 3. Results

### 3.1. Sensitivity and Specificity of the Pan-Serotype FMDV Lateral Flow Strip Test

Purified FMDV was used for the optimization and preliminary assessment of the sensitivity of the LFST. Representative subtypes of each FMDV serotype were detected by the LFST (Figure 1). Furthermore, swine vesicular disease (SVDV) and virus-free buffer were negative on the test line (Figure 1).

Next, 24 epithelial tissue suspensions from naïve animals were tested to determine the diagnostic specificity (DSp) of the LFST, and all samples were negative, returning 100% DSp (data not shown). The ability of the LFST to detect different subtypes of the seven FMDV serotypes was examined. The LFST detected a total of 66 FMDV isolates (17 serotype O, 15 serotype A, 10 serotype Asia1, 7 serotype SAT1, 12 serotype SAT2, 5 serotype SAT3 and 1 serotype C) in the cell culture supernatants, confirming that the test could detect all 7 FMDV serotypes (Figure 2).

The analytical sensitivity of the LFST was evaluated and compared to the AgELISA by testing two-fold serial dilutions of isolates of six FMDV serotypes (excluding C) in culture supernatants. The color intensity on the test line of the LFST was dose-dependent, progressively declining from strong to weak positive to negative for all serotypes (Table 1). The limit of detection (LOD) was identical for the LFST and AgELISA for serotypes SAT1, A and SAT3 at 5.4, 4.8 and 3.7 log_10_ 50% tissue culture infectious dose (TCID_50_)/0.1 mL, respectively. The scoring of the LFST band intensity is shown in Figure 2, and for the AgELISA, an OD ≥ 0.1 is considered a positive result. For serotype O, the LOD for the LFST was 5.1 log_10_ TCID_50_/0.1 mL lower than that for the AgELISA, which was 6.0 log_10_ TCID_50_/0.1 mL. Similarly, for serotype SAT2, the LOD for the LFST was 4.3 log_10_ TCID_50_/0.1 mL, while that for the AgELISA was 4.6 log_10_ TCID_50_/0.1 mL. On the other hand, the LOD for the Asai1 LFST (4.0 log_10_ TCID_50_/0.1 mL) was higher than that of the AgELISA (3.4 log_10_ TCID_50_/0.1 mL). Overall, the analytical sensitivity of the LFST was comparable to that of the AgELISA (Table 1). 

### 3.2. Antigen Detection in Clinical Samples Using the Pan-Serotype FMDV Lateral Flow Strip Test

Tissues collected from animals experimentally inoculated and confirmed as positive for FMDV by RRT-PCR were tested on the pan-serotype FMDV LFST and AgELISA to evaluate whether the LFST can detect FMDV in tissue samples. All tissues were positive by both the LFST and AgELISA (Table 2). 

## 4. Discussion

Previously reported LFSTs for FMDV antigen detection relied on a combination of monoclonal antibodies [11,12,13,14,15,16]. In this report, we successfully developed an LFST using RBIαvβ6, which is a recombinant version of the protein used by all naturally occurring serotypes of FMDV to enter cells [18,19,21,22], and a MAb that recognizes all seven serotypes of FMDV. Indeed, this pan-serotype LFST detected all serotypes of FMDV without cross-reactivity with other vesicular disease viruses, achieving a diagnostic specificity of 100%. Furthermore, with vesicular fluid, epithelial tags from ruptured vesicles or swabs of fresh lesions, the positive detection rate between the LFST and molecular assays is comparable [3,18]. These results render this LFST highly valuable for the rapid detection of FMDV in animals showing typical clinical signs. We anticipate that this test could be performed in the field for early detection and initiation of control measures while samples are being sent to the laboratory for further testing. Molecular assays such as rRT-PCR and RT-LAMP targeting the 3D gene continue to be the most analytically sensitive and specific pan-serotype assays for FMDV [4,6,25,26,27]. However, as opposed to the LFST, field deployable versions of these molecular assays still require expensive equipment and operator training [3,4,5,6]. Therefore, the LFST can be a useful support tool for enforcement of control measures in the face of an active outbreak. Indeed, during the 2007 FMD outbreak in the United Kingdom, the LFST was one of the tests used for FMD diagnosis [28].

On the other hand, there is merit to having a serotype-specific FMDV LFST, especially in endemic pools where circulating serotypes have been well characterized. Detecting FMD and simultaneously identifying the serotype of FMDV responsible for an outbreak allow for early consideration of vaccine choices. However, despite knowing the serotype, vaccine matching is essential to identify the best vaccine against the outbreak [29]. 

Some laboratories in developing countries rely solely on AgELISA for the detection of FMD. These ELISAs are serotype-specific and relatively difficult and require trained technicians and laboratory equipment. We have demonstrated that the sensitivity of this pan-serotype LFST is comparable to the AgELISA for the respective FMDV serotypes. 

Acid treatment of FMDV-positive LFSTs is increasingly used as a safe means of transportation of inactivated FMDV from the field to the laboratory and/or from endemic countries to reference laboratories for further analysis. FMDV sequences have been derived from samples transported in this way, and in some cases, a live virus has been recovered through transfection of nucleic acid from the LFST into cells [30,31]. The pan-serotype LFST can provide a universal vehicle for all serotypes of FMDV, instead of having multiple serotype-specific LFSTs for pools with co-circulation of multiple FMDV serotypes.

## 5. Conclusions

This pan-serotype LFST is a potentially useful tool for the rapid detection of FMD in the field and laboratory, irrespective of the FMDV serotype. The binding characteristics of the recombinant bovine integrin also render the LFST a universal capture and safe transport tool for all serotypes of FMDV following acid inactivation of the membrane. However, additional field evaluation in multiple endemic pools is required. Furthermore, a field-ready version that does not require a cold chain will be ideal for remote settings.

## Figures and Tables

**Figure 1 viruses-14-00785-f001:**
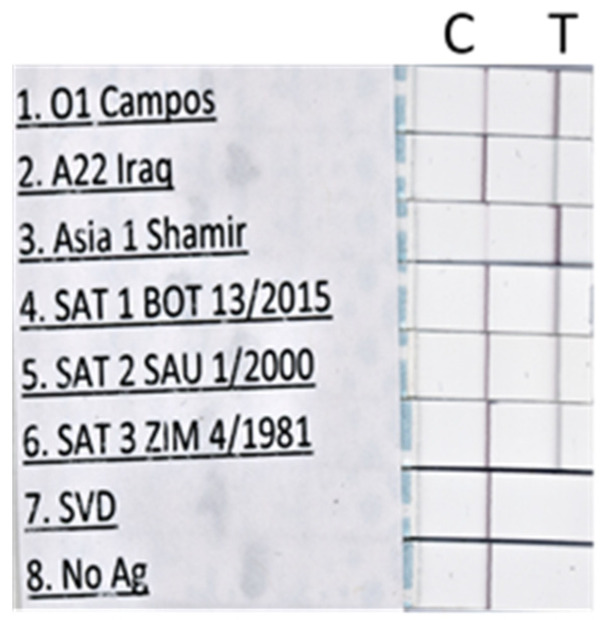
Samples (representative of six foot-and-mouth disease virus serotypes and swine vesicular disease virus) were each mixed with biotinylated recombinant bovine integrin αvβ6 and a gold-conjugated monoclonal antibody (mAb F21-42) in running buffer. A gRAD strip per sample was dipped into the tube containing this mixture which ran through the strip by capillary action, and the results were determined after 10–30 min. A positive result is indicated by visible bands on both the test line (T) and the control line (C). A negative result is indicated by a visible band on the control line only.

**Figure 2 viruses-14-00785-f002:**
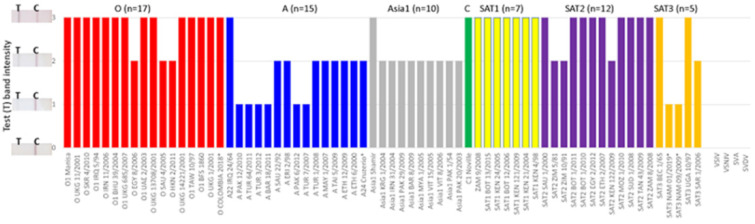
Pan-serotype FMDV strip test results for FMDV and other vesicular disease viruses in cell culture supernatants. Samples were each mixed with biotinylated recombinant bovine integrin αvβ6 and a gold-conjugated monoclonal antibody (mAb F21-42) in running buffer. A gRAD strip per sample was dipped into the tube containing this mixture which ran through the strip by capillary action, and the results were determined after 10–30 min. A positive result is indicated by visible bands on both the test line (T) and the control line (C). A negative result is indicated by a visible band on the control line only. A score of 0, 1, 2 or 3 was recorded for a negative, weak, medium or strong positive test (T) result (band intensity), respectively. An image of a strip test band intensity is shown besides the corresponding numerical score on the Y axis.

**Table 1 viruses-14-00785-t001:** Comparison of analytical sensitivities of antigen detection ELISA (AgELISA) and pan-serotype FMDV lateral flow strip test (LFST) for isolates representing six FMDV serotypes.

	O UKG11/2001			A24 Cruzerio			Asia1 Shamir			SAT1 KEN4/98			SAT2 ZIM10/91			SAT3 ZIM4/81		
Dilution	Titer (Log_10_ TCID_50_/0.1 mL)	AgELISA	LFST	Titer (Log_10_ TCID_50_/0.1 mL)	AgELISA	LFST	Titer (Log_10_ TCID_50_/0.1 mL)	AgELISA	LFST	Titer (Log_10_ TCID_50_/0.1 mL)	AgELISA	LFST	Titer (Log_10_ TCID_50_/0.1 mL)	AgELISA	LFST	Titer (Log_10_ TCID_50_/0.1 mL)	AgELISA	LFST
Neat	6.9	0.60	+++	6.9	2.46	+++	5.8	3.66	+++	6.9	1.77	+++	5.8	0.69	++	5.8	2.41	+++
1:2	6.6	0.42	+++	6.6	2.26	+++	5.5	3.58	+++	6.6	1.26	+++	5.5	0.57	++	5.5	2.18	+++
1:4	6.3	0.28	++	6.3	1.91	+++	5.2	3.23	+++	6.3	0.83	++	5.2	0.41	++	5.2	1.72	+++
1:8	6.0	0.16	++	6.0	1.36	++	4.9	2.31	++	6.0	0.49	++	4.9	0.25	++	4.9	1.27	+++
1:16	5.7	0.09	++	5.7	0.94	++	4.6	1.33	++	5.7	0.26	+	4.6	0.13	+	4.6	0.82	++
1:32	5.4	0.02	+	5.4	0.54	+	4.3	0.73	+	5.4	0.13	+	4.3	0.06	+	4.3	0.47	++
1:64	5.1	0.02	+	5.1	0.30	+	4.0	0.37	+	5.1	0.05	−	4.0	0.04	−	4.0	0.25	+
1:128	4.8	0.02	−	4.8	0.16	+	3.7	0.19	−	4.8	0.03	−	3.7	0.00	−	3.7	0.12	+
1:256	4.5	0.02	−	4.5	0.07	−	3.4	0.11	−	4.5	0.02	−	3.4	0.01	−	3.4	0.055	−
1:512	4.2	0.01	−	4.2	0.04	−	3.1	0.06	−	4.2	0.00	−	3.1	0.01	−	3.1	0.03	−

Comparison of analytical sensitivities of antigen detection ELISA (AgELISA) and pan-serotype FMDV lateral flow strip test (LFST) for six FMDV serotypes. FMDV isolates in culture supernatants were two-fold serially diluted in PBS. Each sample was tested in parallel using the LFST and the AgELISA. Numbers under AgELISA represent optical density (OD) values. An OD ≥ 0.1 is considered a positive result in the AgELISA. A positive result for the LFST is indicated by visible bands on both the test line (T) and the control line (C). A negative result is indicated by a visible band on the control line only. +++ = strong positive; ++ = medium positive; + = weak positive; − = negative result.

**Table 2 viruses-14-00785-t002:** Pan-serotype FMDV lateral flow strip test (LFST) results for FMDV-positive tissue suspensions.

Serotype/Subtype	Animal ID	Tissue Origin	DPI	rRT-PCR Result (Ct)	AgELISA Result (OD)	LFST Result
O UKG 11/2001	Pig 13	Epithelium	3	12.20	0.666	+++
O UKG 11/2001	Pig 14	Epithelium	3	13.50	0.766	+++
O UKG 11/2001	Pig 15	Epithelium	3	15.33	0.437	++
O UKG 11/2001	Pig 16	Epithelium	3	14.32	0.624	+++
O1 BFS/1860	Pig 41	Foot	8	24.69	0.346	+++
O UKG 11/2001	Pig 59	Epithelium	3	18.76	0.586	+++
O UKG 11/2001	Cattle	Foot (interdigital space)	3	13.31	0.516	++
O UKG 11/2001	Cattle	Foot	n/a	13.27	0.513	+++
O1 Manisa	Cattle	Foot	3	17.39	0.386	+++
A IRN 1/2009	Pig 77	Foot (coronary band)	4	14.47	2.496	+++
A IRN 1/2009	Pig 78	Foot (coronary band)	4	19.22	2.133	+++
A IRN 1/2009	Pig 78	Foot (interdigital space)	4	18.26	1.637	+++
A IRN 1/2009	Pig 79	Foot (interdigital space)	n/a	17.14	2.462	+++
A IRN 1/2009	Pig 80	Foot (coronary band)	4	15.59	2.378	+++
ASIA 1 PAK 20/2003	Pig 1	Foot	3	14.48	2.345	+++
ASIA 1 PAK 20/2003	Pig 2	Foot (coronary band)	4	12.09	3.433	+++
ASIA 1 PAK 20/2003	Pig 3	Foot	4	18.56	1.572	+++
ASIA 1 PAK 20/2003	Pig 4	Foot (coronary band)	4	15.96	3.288	+++
ASIA 1 PAK 20/2003	Pig 4	Foot	3	13.54	2.898	+++
SAT 1 BOT 1/68	Cattle	Tongue epithelium	2	14.06	1.843	+
SAT 1 ZAM 9/2008	Pig 81	Foot (interdigital space)	6	15.89	1.138	+++
SAT 1 ZAM 9/2008	Pig 81	Foot (coronary band)	6	17.04	1.111	+++
SAT 1 ZAM 9/2008	Pig 81	Soft palate	6	17.37	0.811	+++
SAT 1 ZAM 9/2008	Pig 84	Foot (interdigital space)	4	19.18	0.933	+++
SAT 2 EGY 6/2012	Pig 89	Foot (coronary band)	3	18.68	0.000	+++
SAT 2 EGY 6/2012	Pig 91	Foot (interdigital space)	3	25.56	0.322	+
SAT 2 EGY 6/2012	Pig 91	Foot (coronary band)	3	25.14	0.324	++
SAT 2 EGY 6/2012	Pig 92	Snout	3	23.64	0.540	+++
SAT 2 EGY 6/2012	Pig 92	Foot (coronary band)	3	21.58	0.780	+++
SAT 3 ZIM 4/81	Pig 5	Hock tissue	3	21.10	3.074	+++
SAT 3 ZIM 4/81	Pig 8	Hock tissue	3	21.74	3.037	+++
SAT 3 SAR 1/2006	Pig 93	Foot (interdigital space)	3	18.56	1.153	+++
SAT 3 SAR 1/2006	Pig 94	Foot (coronary band)	4	18.82	2.494	+++
SAT 3 SAR 1/2006	Pig 94	Foot (interdigital space)	4	18.29	1.904	+++

Pan-serotype FMDV lateral flow strip test (LFST) results for FMDV-positive tissue suspensions. Tissue suspensions positive for FMDV by real-time reverse transcription polymerase reaction (RRT-PCR) were each tested in parallel using the LFST and the AgELISA. An OD ≥ 0.1 is considered a positive result in the AgELISA. A positive result for the LFST is indicated by visible bands on both the test line (T) and the control line (C). A negative result is indicated by a visible band on the control line only. +++ = strong positive; ++ = medium positive; + = weak positive.

## Data Availability

Not applicable.

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
