# Peer review of "Combining a Universal Capture Ligand and Pan-Serotype Monoclonal Antibody to Develop a Pan-Serotype Lateral Flow Strip Test for Foot-and-Mouth Disease Virus Detection"

_viruses, 2022, doi:10.3390/v14040785_

Round 1
Reviewer 1 Report
The development of a strip test (LFST) using recombinant bovine integrin to detect FMDV is the need of the hour. That to if it detects all 7 FMDV serotypes with sensitivity is welcomed. This test will be useful in endemic settings also. The paper is well written, Recommended for publication.
Author Response
The development of a strip test (LFST) using recombinant bovine integrin to detect FMDV is the need of the hour. That to if it detects all 7 FMDV serotypes with sensitivity is welcomed. This test will be useful in endemic settings also. The paper is well written, Recommended for publication.
No change is required from reviewer #1 .
Reviewer 2 Report
Comments for the Author:
Foot-and-mouth disease virus (FMDV) is a highly contagious virus affecting cloven-hoofed animals, including cattle, pigs, and sheep. Rapid detection of pan-serotype FMDV is critical to limit the devastating economic losses due to FMD. In this work, the authors developed a pan-serotype lateral-flow strip test (LFST) that uses recombinant bovine integrin αvβ6 as a universal capture ligand and a pan-serotype monoclonal antibody (mAb) to detect all 7 FMDV serotypes. Overall, this is an advance in FMDV detection, which could be used to detect FMD in the field and laboratory rapidly.
The methods and the data presented look convincing and well-presented. However, some points are needed to be addressed before publication.
I recommend the following changes:
Major concern:
My main issue is that this paper uses recombinant bovine integrin αvβ6. I wonder whether the LFST can detect FMDV isolated from goats and sheep?
Minor issues:
1) In the “2.1. Production and Biotinylation of recombinant bovine integrin αvβ6”, 25ml, 0.875ml, 1mg/ml and NaN3 should be changed to 25 ml, 0.875 ml, 1mg /ml and NaN3, and I suggest you check the spelling throughout the manuscript.
2) You mentioned the analytical sensitivity of the LFST was comparable to that of AgELISA (Table 1). More specific descriptions are needed, including the cut-offs, the lowest detection limits of both methods.
Author Response
Reviewer’s comment: My main issue is that this paper uses recombinant bovine integrin αvβ6. I wonder whether the LFST can detect FMDV isolated from goats and sheep?
Response: This integrin can recognize FMDv from sheep, goats and other species as demonstrated by Ferris et al. (Validation of a recombinant integrin alphavbeta6/monoclonal antibody based antigen elisa for the diagnosis of foot-and-mouth disease. J Virol Methods 2011, 175, 253-260).
This information has been included in the manuscript.
Minor issues:
Reviewer’s comment: 1) In the “2.1. Production and Biotinylation of recombinant bovine integrin αvβ6”, 25ml, 0.875ml, 1mg/ml and NaN3 should be changed to 25 ml, 0.875 ml, 1mg /ml and NaN3, and I suggest you check the spelling throughout the manuscript.
Response: The suggested corrections have been made.
Reviewer’s comment 2) You mentioned the analytical sensitivity of the LFST was comparable to that of AgELISA (Table 1). More specific descriptions are needed, including the cut-offs, the lowest detection limits of both methods.
Response:
The following has been added to the applicable section. “The limit of detection (LOD) was identical for LFST and AgELISA for serotypes SAT1, A and SAT3 at 5.4, 4.8 and 3.7 log10 50 % tissue culture infectious dose (TCID50)/0.1 ml respectively. The scoring of LFST band intensity is shown in Figure 2 and for the AgELISA an O.D. ≥ 0.1 is considered a positive result. For serotype O, the LOD for LFST was 5.1 log10 TCID50/0.1 ml lower than that for AgELISA which was 6.0 log10 TCID50/0.1 ml. Similarly, for serotype SAT2 the LOD for LFST was 4.3 log10 TCID50/0.1 ml while that for AgELISA was 4.6 log10 TCID50/0.1 ml. On the other hand, the LOD for Asai1 LFST (4.0 log10 TCID50/0.1 ml) was higher than that of the AgELISA (3.4 log10 TCID50/0.1 ml). Overall, the analytical sensitivity of the LFST was comparable to that of AgELISA (Table 1).”